Extended Abstract Track

# Surfing on the Neural Sheaf

**Julian Suk**[*]                                                          J.M.SUK@UTWENTE.NL
*University of Twente*

**Lorenzo Giusti**[*]                                          LORENZO.GIUSTI@UNIROMA1.IT
*Sapienza University of Rome*

**Tamir Hemo**[*]                                                   THEMO@CALTECH.EDU
*California Institute of Technology*

**Miguel Lopez**                                                MLOPEZ3@SAS.UPENN.EDU
*University of Pennsylvania*

**Konstantinos Barmpas**          KONSTANTINOS.BARMPAS16@IMPERIAL.AC.UK
*Imperial College London*

**Cristian Bodnar**                                                    CB2015@CAM.AC.UK
*University of Cambridge*

**Editors:** Sophia Sanborn, Christian Shewmake, Simone Azeglio, Arianna Di Bernardo, Nina Miolane

## Abstract

The deep connections between Partial Differential Equations (PDEs) and Graph Neural Networks (GNNs) have recently generated a lot of interest in PDE-inspired architectures for learning on graphs. However, despite being more interpretable and better understood via well-established tools from PDE analysis, the dynamics these models use are often too simple for complicated node classification tasks. The recently proposed Neural Sheaf Diffusion (NSD) models address this by making use of an additional geometric structure over the graph, called a sheaf, that can support a provably powerful class of diffusion equations. In this work, we propose Neural Sheaf Propagation (NSP), a new PDE-based Sheaf Neural Network induced by the wave equation on sheaves. Unlike diffusion models that are characterised by a dissipation of energy, wave models conserve a certain energy, which can be beneficial for node classification tasks on heterophilic graphs. In practice, we show that NSP obtains competitive results with NSD and outperforms many other existent models on several datasets.

**Keywords:** Graph Neural Networks, Sheaf Theory, Partial Differential Equations

## 1. Introduction

Since their introduction (Scarselli et al., 2008), Graph Neural Networks (GNNs) have shown outstanding results in a broad range of applications, from *physics simulations* (Shlomi et al., 2020) to *protein folding* (Jumper et al., 2021). The key idea is to leverage the inductive bias induced by the topology of graph-structured data, represented by the connectivity structure, to perform graph representation learning tasks. However, it has been shown that pioneering works on GNNs, like the Graph Convolutional Networks (GCNs) (Kipf and Welling, 2016) tend to suffer from oversmoothing Oono and Suzuki (2019) and perform poorly on heterophilic datasets Zhu et al. (2020a). In the former case, GCNs with many

---

[*] Equal contribution

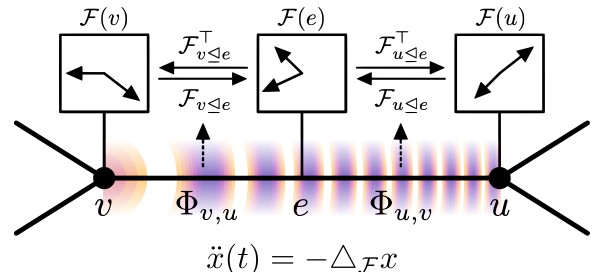

Figure 1: Neural sheaf propagation (NSP) induced by the wave equation on sheaves. Within the sheaf framework, we can use the (normalised) sheaf Laplacian $\triangle_{\mathcal{F}}$ to discretise the solution to the wave equation over nodes of the graph which forms our network layers.

layers tend to have nodes that aggregate features from remote parts of the underlying graph, leading to the inability to exhibit local patterns over the underlying graph domain, which jeopardises the overall performance. In the latter case, GCNs assume that nodes, that are connected, will share the same label, which is an inhibitive inductive bias in most of the real-world scenarios.

To overcome these limitations, Hansen and Gebhart (2020), Bodnar et al. (2022) introduced generalisations of graph convolutional networks by equipping the underlying graph with the structure of a cellular sheaf Curry (2014) (see section 2 for a review). Sheaves are used to construct convolution via the *sheaf laplacian*. The richer harmonic space of such sheaf laplacians give better node separation powers. Hansen and Gebhart (2020) introduced sheaf convolutional networks with a one dimensional sheaf, while Bodnar et al. (2022) introduced *Neural Sheaf Diffusion* which allows higher dimensional sheaves in combination with learning the sheaf itself from the data Barbero et al. (2022); Bodnar et al. (2022).

In this work, we aim to exploit recent works on PDE-based GNNs Eliasof et al. (2021); Chamberlain et al. (2021) to introduce *Neural Sheaf Propagation* (NSP), a procedure used to induce the wave equation on sheaves. Unlike sheaf diffusion models Bodnar et al. (2022); Barbero et al. (2022) or other ODE- or PDE-based GNNs Di Giovanni et al. (2022); Chamberlain et al. (2021), characterised by a dissipation of energy, wave models conserve energy. This can be beneficial for node classification tasks on heterophilic graphs since the features will generally tend to be less smooth. In related work, GNNs that are discretisations of second-order ODEs have been shown to avoid oversmoothing Rusch et al. (2022).

In Section 2, we review the required background on graph neural networks, cellular sheaf theory and neural sheaf diffusion. Section 3 defines the Neural Sheaf Propagation (NSP), a procedure used to induce the wave equation on sheaves. Finally, in Section 4, evaluate this technique on several datasets with varying homophily levels for the node classification task.

## 2. Background

We encourage the reader to consult Curry (2014); Hansen and Ghrist (2019); Hansen (2020) for an in-depth overview of cellular sheaf theory and Bodnar et al. (2022) for a theoretical analysis regarding neural sheaf diffusion.

Extended Abstract Track

**Cellular Sheaves** Given an *undirected* graph $G = (V, E)$, a *cellular sheaf* $\mathcal{F}$ over $G$ consists of: (1) an assignment of a real vector space $\mathcal{F}(v)$ for each $v \in V$, (2) an assignment of a real vector space $\mathcal{F}(e)$ for each $e \in E$, and (3) a linear map $\mathcal{F}_{v \trianglelefteq e} : \mathcal{F}(v) \to \mathcal{F}(e)$ whenever $e$ is an edge containing $v$.

We refer to the vector spaces $\mathcal{F}(v)$ and $\mathcal{F}(e)$ as *stalks* and the linear maps between them as *restriction maps*. For our purposes, elements of a vertex stalk $\mathcal{F}(v)$ correspond to feature vectors $x_v$, while the edge stalks $\mathcal{F}(e)$ serve as ancillary spaces in which vertex features can mix. The space $C^0(G; \mathcal{F})$ of *0-cochains* consists of collections of feature vectors $\boldsymbol{x} = (x_v)_{v \in V}$. The *sheaf Laplacian* $L_{\mathcal{F}} : C^0(G, \mathcal{F}) \to C^0(G, \mathcal{F})$ is a linear operator defined on each stalk as $L_{\mathcal{F}}(\boldsymbol{x})_v = \sum_{u, v \trianglelefteq e} \mathcal{F}_{v \trianglelefteq e}^T (\mathcal{F}_{v \trianglelefteq e} x_v - \mathcal{F}_{u \trianglelefteq e} x_u)$ The sheaf Laplacian is a symmetric positive semi-definite block matrix. The diagonal blocks are $L_{\mathcal{F}_{v,v}} = \sum_{v \trianglelefteq e} \mathcal{F}_{v \trianglelefteq e}^\top \mathcal{F}_{v \trianglelefteq e}$, while the off-diagonal blocks are $L_{\mathcal{F}_{v,u}} = -\mathcal{F}_{v \trianglelefteq e}^\top \mathcal{F} u \trianglelefteq e$. The *normalised sheaf Laplacian* $\Delta_{\mathcal{F}}$ is defined as $\Delta_{\mathcal{F}} = D^{-\frac{1}{2}} L_{\mathcal{F}} D^{-\frac{1}{2}}$ where $D$ is the block-diagonal of $L_{\mathcal{F}}$.

**Neural Sheaf Diffusion** Let us consider a graph $G = (V, E)$. Consider also $x \in C^0(G, \mathcal{F})$ to be an *nd*-dimensional vector obtained by stacking vertically $n$ individual $d$-dimensional feature vectors $\mathbf{x}_v \in \mathcal{F}(v)$ equipped to each node $v \in V$. If each feature vector allows a number of channels equal to $f$, the result is a feature matrix $X \in \mathbb{R}^{(nd) \times f}$ such that each column of $\mathbf{X}$ is a vector that belongs in $C^0(G, \mathcal{F})$. *Sheaf diffusion* is a process defined by the following differential equation:

$$\dot{\mathbf{X}}(t) = -\Delta_{\mathcal{F}(t)} \mathbf{X}(t), \tag{1}$$

evolving on $(G, \mathcal{F})$. In Bodnar et al. (2022), the authors generated a neural network with layers based on (1). More precisely, the model has layers of the form:

$$\mathbf{X}_{t+1} = \mathbf{X}_t - \sigma(\Delta_{\mathcal{F}(t)}(\mathbf{I} \otimes W_1^t)\mathbf{X}_t W_2^t) \tag{2}$$

where $\sigma$ is an activation function. It is worth to remark that both the sheaf $\mathcal{F}(t)$ and the weights $\mathbf{W}_1^t, \mathbf{W}_2^t$ in equation (2) vary in time, implying that the underlying "geometry" is dynamic as well. The sheaf $\mathcal{F}(t)$ is learned as a function of the data $\mathbf{X}(t)$.

## 3. Neural Sheaf Propagation

The interpretation of message passing convolution networks as diffusion processes opens the question of building models corresponding to other dynamical systems. In this work we focus on the dynamics induced by a hyperbolic PDE, the wave equation. We consider a time-dependent process $X(t) \in C^0(G; \mathcal{F})$. The wave equation is given by:

$$\ddot{\mathbf{X}}(t) = -\Delta_{\mathcal{F}(t)} \mathbf{X}(t). \tag{3}$$

Unlike the diffusive dynamics of the heat equation, the wave equation conserves a certain total energy of the signal. Namely, with the energy defined by $\mathcal{E}_{\mathcal{F}}(\mathbf{X}) = \frac{1}{2}\left(||\dot{\mathbf{X}}||^2 + \mathbf{X}^T \Delta_{\mathcal{F}(t)} \mathbf{X}\right)$, we have the following result:

**Proposition 1** *The propagation through the PDE in (3) preserves the energy $\mathcal{E}_{\mathcal{F}}$.*

**Proof** The proof is standard. Taking the derivative of the energy, we get:

$$\dot{\mathcal{E}}_{\mathcal{F}}(\mathbf{X}) = \dot{\mathbf{X}}^T \ddot{\mathbf{X}} + \dot{\mathbf{X}}^T \Delta_{\mathcal{F}(t)} \mathbf{X} = \dot{\mathbf{X}}^T (\ddot{\mathbf{X}} + \Delta_{\mathcal{F}(t)} \mathbf{X}) = 0$$

with the last equality coming from (3). ■

Motivated by this, we introduce a model we call **Neural Sheaf Propagation**. It is a discretisation of a general hyperbolic version of (3), followed by a non-linearity. Namely, we replace the Laplacian term by $\Delta_{\mathcal{F}(t)}(\mathbf{I} \otimes W_1^t)\mathbf{X}_t W_2^t$, and add an activation term.

The discretisation is done using the leapfrog method. Put together, the layers of the network have the structure:

$$\mathbf{X}_{t+1} = 2\mathbf{X}_t - \mathbf{X}_{t-1} - \sigma(\Delta_{\mathcal{F}(t)}(\mathbf{I} \otimes W_1^t \mathbf{X}_t W_2^t))$$

## 4. Experiments

We evaluate the model proposed in this work on several different datasets, comparing its performance against several models present in the graph representation learning literature. Similar to NSD, our model comes with a few variations depending on the type of restriction maps that it learns: diagonal, orthogonal and general matrices. The results are shown in Table 1.

The datasets are sorted based on their homophily coefficient, the fraction of edges which connect nodes of the same class label which ranges from 0.11 to 0.81. The higher the homophily coefficient, the more homophilic is the datasets. The results are collected over 10 fixed splits, where 48%, 32%, and 20% of nodes per class are used for training, validation, and testing, respectively. The results shown in Table 1 are selected by taking the test accuracy corresponding to the highest validation accuracy.

## 5. Conclusion

In this work we proposed Neural Sheaf Propagation (NSP), a novel PDE-based architecture induced by the wave equation on (cellular) sheaves. The purpose of this work was to leverage the recent development on Neural Sheaf Diffusion (NSD) from Bodnar et al. (2022) to build a model to perform convolution operations that preserve the overall energy of the system rather than dispel it through the diffusion process. Evaluation of the proposed method shows that this technique achieves competitive results in several node classification tasks and outperforms the other models in the most heterophilic setting. However, the presented results are preliminary. We plan to investigate the interplay between energy conservation and classification accuracy on heterophilic graphs more systematically in the future.

### Acknowledgments

We are grateful to all those people involved in the 2022 London Geometry and Machine Learning Summer School (LOGML), where this research project started. In particular, we would like to thank the organising committee: Josh Southern, Michelle Li, Tim King, Sara Veneziale, and Francesco Viganò.

# Extended Abstract Track

Table 1: Results on node classification datasets sorted by their homophily level. Top three models are coloured by **First**, **Second**, **Third**. Our models are marked with **NSP** for Neural Sheaf Propagation. NSD represents Neural Sheaf Diffusion

| | Texas | Wisconsin | Film | Squirrel | Chameleon | Cornell | Citeseer | Pubmed | Cora |
|---|---|---|---|---|---|---|---|---|---|
| Hom level | **0.11** | **0.21** | **0.22** | **0.22** | **0.23** | **0.30** | **0.74** | **0.80** | **0.81** |
| #Nodes | 183 | 251 | 7,600 | 5,201 | 2,277 | 183 | 3,327 | 18,717 | 2,708 |
| #Edges | 295 | 466 | 26,752 | 198,493 | 31,421 | 280 | 4,676 | 44,327 | 5,278 |
| #Classes | 5 | 5 | 5 | 5 | 5 | 5 | 7 | 3 | 6 |
| **Diag-NSP** | 85.68±5.93 | 89.02±3.84 | 37.12±1.31 | 48.78±2.45 | 61.80±2.31 | 76.22±4.49 | 76.82±1.78 | 89.38±0.56 | 87.02±1.91 |
| **O(d)-NSP** | 87.03±5.51 | 87.06±4.13 | 36.56±1.15 | 49.54±1.87 | 61.01±4.14 | 76.22±4.95 | 76.77±1.61 | 89.23±0.51 | 86.22±1.55 |
| **Gen-NSP** | 84.60±3.43 | 87.45±4.40 | 37.07±1.20 | 50.11±2.03 | 62.85±1.98 | 76.49±5.28 | 76.85±1.48 | 89.42±0.33 | 87.38±1.14 |
| NSD (max) | 85.95±5.51 | 89.41±4.74 | 37.81±1.15 | 56.34±1.32 | 68.68±1.73 | 86.49±7.35 | 77.14±1.85 | 89.49±0.40 | 87.30±1.15 |
| GGCN | 84.86±4.55 | 86.86±3.29 | 37.54±1.56 | 55.17±1.58 | 71.14±1.84 | 85.68±6.63 | 77.14±1.45 | 89.15±0.37 | 87.95±1.05 |
| H2GCN | 84.86±7.23 | 87.65±4.98 | 35.70±1.00 | 36.48±1.86 | 60.11±2.15 | 82.70±5.28 | 77.11±1.57 | 89.49±0.38 | 87.87±1.20 |
| GPRGNN | 78.38±4.36 | 82.94±4.21 | 34.63±1.22 | 31.61±1.24 | 46.58±1.71 | 80.27±8.11 | 77.13±1.67 | 87.54±0.38 | 87.95±1.18 |
| FAGCN | 82.43±6.89 | 82.94±7.95 | 34.87±1.25 | 42.59±0.79 | 55.22±3.19 | 79.19±9.79 | N/A | N/A | N/A |
| MixHop | 77.84±7.73 | 75.88±4.90 | 32.22±2.34 | 43.80±1.48 | 60.50±2.53 | 73.51±6.34 | 76.26±1.33 | 85.31±0.61 | 87.61±0.85 |
| GCNII | 77.57±3.83 | 80.39±3.40 | 37.44±1.30 | 38.47±1.58 | 63.86±3.04 | 77.86±3.79 | 77.33±1.48 | 90.15±0.43 | 88.37±1.25 |
| Geom-GCN | 66.76±2.72 | 64.51±3.66 | 31.59±1.15 | 38.15±0.92 | 60.00±2.81 | 60.54±3.67 | 78.02±1.15 | 89.95±0.47 | 85.35±1.57 |
| PairNorm | 60.27±4.34 | 48.43±6.14 | 27.40±1.24 | 50.44±2.04 | 62.74±2.82 | 58.92±3.15 | 73.59±1.47 | 87.53±0.44 | 85.79±1.01 |
| GraphSAGE | 82.43±6.14 | 81.18±5.56 | 34.23±0.99 | 41.61±0.74 | 58.73±1.68 | 75.95±5.01 | 76.04±1.30 | 88.45±0.50 | 86.90±1.04 |
| GCN | 55.14±5.16 | 51.76±3.06 | 27.32±1.10 | 53.43±2.01 | 64.82±2.24 | 60.54±5.30 | 76.50±1.36 | 88.42±0.50 | 86.98±1.27 |
| GAT | 52.16±6.63 | 49.41±4.09 | 27.44±0.89 | 40.72±1.55 | 60.26±2.50 | 61.89±5.05 | 76.55±1.23 | 87.30±1.10 | 86.33±0.48 |
| MLP | 80.81±4.75 | 85.29±3.31 | 36.53±0.70 | 28.77±1.56 | 46.21±2.99 | 81.89±6.40 | 74.02±1.90 | 87.16±0.37 | 75.69±2.00 |

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

# Extended Abstract Track

Petar Velickovic, Guillem Cucurull, Arantxa Casanova, Adriana Romero, Pietro Lio, and Yoshua Bengio. Graph attention networks. *stat*, 1050:20, 2017.

Yujun Yan, Milad Hashemi, Kevin Swersky, Yaoqing Yang, and Danai Koutra. Two sides of the same coin: Heterophily and oversmoothing in graph convolutional neural networks. *arXiv preprint arXiv:2102.06462*, 2021.

Lingxiao Zhao and Leman Akoglu. Pairnorm: Tackling oversmoothing in gnns. *arXiv preprint arXiv:1909.12223*, 2019.

Jiong Zhu, Yujun Yan, Lingxiao Zhao, Mark Heimann, Leman Akoglu, and Danai Koutra. Beyond homophily in graph neural networks: Current limitations and effective designs. *Advances in Neural Information Processing Systems*, 33:7793–7804, 2020a.

Jiong Zhu, Yujun Yan, Lingxiao Zhao, Mark Heimann, Leman Akoglu, and Danai Koutra. Generalizing graph neural networks beyond homophily. *arXiv preprint arXiv:2006.11468*, 2020b.

## Appendix A. Details on experiments

Table 1 contains accuracy results for a wide range of models, along with ours, Conn-NSD, for node classification tasks. The GNN models in Table 1 can be clustered in three main categories: (1) Classical: GCN Kipf and Welling (2016), GAT Velickovic et al. (2017), GraphSAGE Hamilton et al. (2017); (2) Models for heterophilic settings: GGCN Yan et al. (2021), Geom-GCN Pei et al. (2020), H2GCN Zhu et al. (2020b), GPRGNN Chien et al. (2020), FAGCN Bo et al. (2021), MixHop Abu-El-Haija et al. (2019); (3) Models which address over-smoothing: GCNII Chen et al. (2020), PairNorm Zhao and Akoglu (2019). To have a baseline for this learning task a Multi-Layer Perceptron (MLP) is also taken into account, whose result reported in the last row of Table 1. The MLP is trained using only the node features and it used to quantify the relevance of the inductive bias provided by graph structure in performing the learning task.

Table 2 contains the ranges of hyperparameters used in the experiments.

Table 2: Hyper-parameter ranges for the experiments.

| | **WebKB** | **Wikipedia** | **Planetoid** | **Film** |
|---|---|---|---|---|
| Hidden channels | $\{8, 16, 32\}$ | $\{8, 16, 32, 64\}$ | $\{8, 16, 32, 64\}$ | $\{8, 16, 32\}$ |
| Stalk dim $d$ | $[2, 5]$ | $[2, 5]$ | $[2, 5]$ | $[2, 5]$ |
| Layers | $\{2, 3, 4, 5, 6\}$ | $\{2, 4, 6, 8\}$ | $\{2, 4, 6, 8\}$ | $\{1, 2, 3\}$ |
| Learning rate | $[1 \times 10^{-3}, 1 \times 10^{-1}]$ | $[1 \times 10^{-3}, 1 \times 10^{-1}]$ | $[1 \times 10^{-3}, 1 \times 10^{-1}]$ | $[1 \times 10^{-3}, 1 \times 10^{-1}]$ |
| Weight decay (regular parameters) | Log-U$(-9.2, -4.5)$ | Log-U$(-10.0, -4.5)$ | Log-U$(-9.2, -4.5)$ | Log-U$(-9.2, -4.5)$ |
| Weight decay (sheaf parameters) | Log-U$(-11.0, -4.5)$ | Log-U$(-11.0, -4.5)$ | Log-U$(-11.0, -4.5)$ | Log-U$(-11.0, -4.5)$ |
| Input dropout | range(0.1, 0.9, 0.1) | range(0.1, 0.9, 0.1) | range(0.1, 0.9, 0.1) | - |
| Layer dropout | range(0.1, 0.9, 0.1) | range(0.1, 0.9, 0.1) | range(0.1, 0.9, 0.1) | - |
| Step Size | $[0.1, 1.0]$ | $[0.1, 1.0]$ | $[0.1, 1.0]$ | $[0.1, 1.0]$ |
| Use Second Linear Transform | - | $\{0, 1\}$ | $\{0, 1\}$ | $\{0, 1\}$ |
| Use Higher P | $\{0, 1\}$ | $\{0, 1\}$ | $\{0, 1\}$ | $\{0, 1\}$ |
| Use Lower P | $\{0, 1\}$ | $\{0, 1\}$ | $\{0, 1\}$ | $\{0, 1\}$ |
| # $\triangle$ eigenvectors | $\{2, 8, 16\}$ | - | - | |
| New $\triangle$ each step | $\{0, 1\}$ | - | - | |
| Optimiser | Adam | Adam | Adam | Adam |

