# OpenReview forum: "Surfing on the Neural Sheaf"
_NeurIPS.cc/2022/Workshop/NeurReps — NeurReps 2022 Poster_

### Official Review · Reviewer_wdrk · 2022-10-15
**Interesting abstract**

**Confidence:** 4
**Soundness:** 3
**Presentation:** 4
**Contribution:** 3
**Overall Rating:** 6

**Summary:**

The paper introduces the Neural Sheaf Propagation (NSP), a novel PDE-based architecture induced by the wave equation on (cellular) sheaves. The advantage of this model is that it performs convolution operations that preserve the overall energy of the system rather than dispel it through the diffusion process. This property is expected to be beneficial in heterophilic datasets.

**Questions:**

It is not clear what is the advantage of this approach with respect to the many other different approaches that are proposed in the same broader area.

How does the nonlinearity, and the time dependent update of the weights affect the propagation?

**Limitations:**

A more extended version would help understand better the intuition, and the experimental results.

**Recommended Decision:**

3: Accept

**Relevance:**

4: Highly relevant

**Strengths And Weaknesses:**

The abstract introduces one additional contribution in the topic of designing better GNNs architectures based on PDEs. The topic is timely and interesting, and the proposed extension is valid. The experimental results are adequate for an abstract submission and the text is compact and easy to follow.

Some more emphasis should be given on the intuition behind the proposed framework. Given that the architecture is built on updates of the form of (3), how does this update, and in particular the use of the nonlinearity, affect the properties of the diffusion?

**Submission Track:**

Extended Abstract (4 Page)

---

### Official Review · Reviewer_gjDc · 2022-10-15
**Do we really need energy conservation?**

**Confidence:** 5
**Soundness:** 3
**Presentation:** 3
**Contribution:** 2
**Overall Rating:** 6

**Summary:**

Using the same technique that turns the heat equation into *Neural Sheaf Diffusion* (NSD) (Bodnar et al., 2022), the authors transform the wave equation into a graph neural network called *Neural Sheaf Propagation* (NSP). Unlike NSD, which dissipates the "energy" like the heat equation, NSP conserves the energy like the wave equation; this allows NSP to have more predictive power on heterophilic graphs (i.e. connected nodes tend to have dissimilar features). Experiments show that NSP performs well on node classification datasets with low homophily levels.


Reference

Cristian Bodnar, Francesco Di Giovanni, Benjamin Paul Chamberlain, Pietro Li`o, and Michael M. Bronstein. Neural sheaf diffusion: A topological perspective on heterophily and oversmoothing in gnns, 2022. URL https://arxiv.org/abs/2202.04579.

**Questions:**

None

**Limitations:**

The authors have not addressed the limitations of NSP.

**Recommended Decision:**

3: Accept

**Relevance:**

4: Highly relevant

**Strengths And Weaknesses:**

# Strengths

* The authors provide crystal clear introduction and background material, with extensive literature review.
* The experiment is extensive as well; it is performed over a large number of models on multiple datasets with various levels of homophily.

# Weaknesses

* Connection between GNN and PDE: as of now, the definition of "energy" in the context of GNN is still unclear. What does it mean for the energy to "dissipate"? What does it mean for the energy to be "preserved"? Understanding GNN's energy might be the key to developing GNNs that can deal hetero philic graphs to a great effect.
* From the experiment, the proposed NSP is not a clear winner over NSD, which casts doubt on whether there is a connection between energy and heterophily at all.

**Submission Track:**

Extended Abstract (4 Page)

---

### Official Review · Reviewer_UeQu · 2022-10-18

**Confidence:** 5
**Soundness:** 3
**Presentation:** 3
**Contribution:** 2
**Overall Rating:** 5

**Summary:**

The authors propose a new GNN model derived from the discretization of the wave equation, which guarantees for the energy of the signal to be preserved.

**Questions:**

Section 1. :"Heterophily" is not a problem in itself, but a property of the data. The problem is the inability of the models to deal with heterohilic data.

Citations should be in parenthesis.

**Limitations:**

Experimental results are extremely poor, which shed doubts on whether this approach works at all. On Chameleon and Squirrel, which are the two most interesting datasets used, the approach vastly underperforms many other methods, including NSD. The authors claim that NSP is suitable for heterophilic data is not sustained by these results.

**Recommended Decision:**

2: Borderline

**Relevance:**

4: Highly relevant

**Strengths And Weaknesses:**

Strengths:
- Nice theoretical idea based on the wave equation which allows to preserve the energy of the signal

Weaknesses:
- More theoretical analysis on why the conservation of the energy is good may be required. Maybe we do want to change it in some cases?
- Poor empirical performance of the model

**Submission Track:**

Extended Abstract (4 Page)

---

### Decision · Program_Chairs · 2022-10-21

Accept (Poster)